# ALIGN WITH PURPOSE: OPTIMIZE DESIRED PROPERTIES IN CTC MODELS WITH A GENERAL PLUG-AND-PLAY FRAMEWORK

**Eliya Segev**[*]   **Maya Alroy**[*]   **Ronen Katsir**   **Noam Wies**   **Ayana Shenhav**   **Yael Ben-Oren**

**David Zar**   **Oren Tadmor**   **Jacob Bitterman**   **Amnon Shashua**   **Tal Rosenwein**

## ABSTRACT

Connectionist Temporal Classification (CTC) is a widely used criterion for training supervised sequence-to-sequence (seq2seq) models. It learns the alignments between the input and output sequences by marginalizing over the perfect alignments (that yield the ground truth), at the expense of the imperfect ones. This dichotomy, and in particular the equal treatment of all perfect alignments, results in a lack of controllability over the predicted alignments. This controllability is essential for capturing properties that hold significance in real-world applications. Here we propose *Align With Purpose (AWP)*, a **general Plug-and-Play framework** for enhancing a desired property in models trained with the CTC criterion. We do that by complementing the CTC loss with an additional loss term that prioritizes alignments according to a desired property. AWP does not require any intervention in the CTC loss function, and allows to differentiate between both perfect and imperfect alignments for a variety of properties. We apply our framework in the domain of Automatic Speech Recognition (ASR) and show its generality in terms of property selection, architectural choice, and scale of the training dataset (up to 280,000 hours). To demonstrate the effectiveness of our framework, we apply it to two unrelated properties: token emission time for latency optimization and word error rate (WER). For the former, we report an improvement of up to 590ms in latency optimization with a minor reduction in WER, and for the latter, we report a relative improvement of 4.5% in WER over the baseline models. To the best of our knowledge, these applications have never been demonstrated to work on this scale of data. Notably, our method can be easily implemented using only a few lines of code[1] and can be extended to other alignment-free loss functions and to domains other than ASR.

# 1 INTRODUCTION

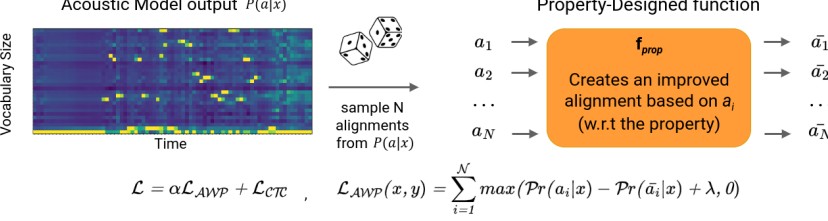

$$\mathcal{L} = \alpha \mathcal{L}_{AWP} + \mathcal{L}_{CTC} \quad , \quad \mathcal{L}_{AWP}(x,y) = \sum_{i=1}^{N} max(\mathcal{P}r(a_i|x) - \mathcal{P}r(\bar{a}_i|x) + \lambda, \, 0)$$

Figure 1: The Align With Purpose flow: $N$ alignments are sampled from the output of a pre-trained CTC model on which $f_{prop}$ is applied to create $N$ pairs of alignments. Then, hinge loss with an adjustable weight is applied on the probabilities of each pair of alignments, trained jointly with a CTC loss. See full details in section 2.2

---

[*]Equally contributed. email: {first.last}@orcam.com
[1]The code will be made publicly available in the supplementary materials.

Sequence-to-sequence (seq2seq) tasks, in which the learner needs to predict a sequence of labels from unsegmented input data, are prevalent in various domains, e.g. handwriting recognition (Graves & Schmidhuber, 2008), automatic speech recognition (Collobert et al., 2016; Hannun et al., 2014), audio-visual speech recognition (Afouras et al., 2018), neural machine translation (Huang et al., 2022), and protein secondary structure prediction (Yang et al., 2022), to name a few. For years, optimizing a seq2seq task required finding a suitable segmentation, which is an explicit alignment between the input and output sequences. This is a severe limitation, as providing such segmentation is difficult (Graves, 2012).

Two main approaches were introduced to overcome the absence of an explicit segmentation of the input sequence, namely soft and hard alignment. Soft alignment methods use attention mechanism (Chan et al., 2016; Vaswani et al., 2017) that softly predict the alignment using attention weights. Hard alignment methods learn in practice an explicit alignment (Graves et al., 2006; Graves, 2012; Collobert et al., 2016), by marginalizing over all alignments that correspond to the ground truth (GT) labels.

As streaming audio and video become prevalent (Cisco, 2018), architectures that can work in a streaming fashion gain attention. Although soft alignment techniques can be applied in chunks for streaming applications (Bain et al., 2023), their implementation is not intuitive and is less computationally efficient compared to hard alignment methods, which are naturally designed for streaming processing. Among the hard alignment methods, the CTC criterion (Graves et al., 2006) is a common choice due to its simplicity and interpretability. During training, CTC minimizes the negative log-likelihood of the GT sequence. To overcome the segmentation problem, CTC marginalizes over all possible input-GT output pairings, termed perfect alignments. This is done using an efficient forward-backward algorithm, which is the core algorithm in CTC.

CTC has a by-product of learning to predict an alignment without direct supervision, as CTC posteriors tend to be peaky (Zeyer et al., 2021; Tian et al., 2022), and hence the posterior of a few specific alignments are dominant over the others. While this implicit learning is useful, it comes at the cost of the inability to control other desired properties of the learned alignment. This can be explained by the inherent dichotomy of the CTC, which leads to a lack of additional prioritization within perfect or imperfect alignments.

However, many real-world seq2seq applications come with a property that can benefit from or even require such prioritization. For example, in the contexts of ASR and OCR, a standard metric to test the quality of a system is the word error rate (WER). Therefore, prioritizing imperfect alignments with low WER can improve the performance of a system measured by this metric, thereby reducing the gap between the training and testing criteria (Graves & Jaitly, 2014). Another example is a low-latency ASR system. Here, even a perfect CTC score can only guarantee a perfect transcription while completely disregard the latency of the system. Clearly, in this setting, for an application that requires fast response, prioritizing alignments with fast emission time is crucial. Figure 2 visualizes the aforementioned properties. In general, there are many other properties that also necessitate prioritization between alignments, whether perfect or imperfect.

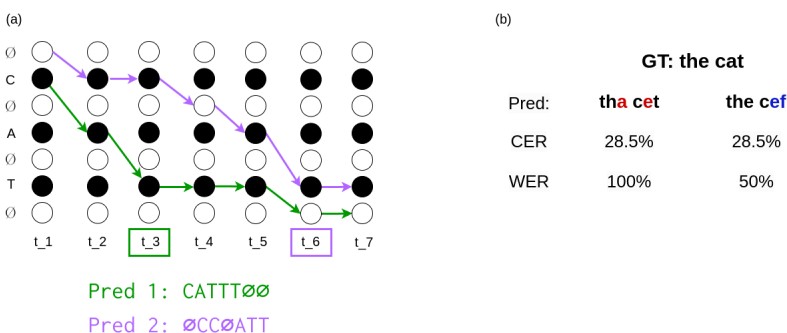

Figure 2: A visualization of two properties that are not captured by CTC. (a) Emission Time: Two alignments that yield the same text, but the green alignment emits the last token of 'CAT' at timestamp 3 (t_3) while the purple alignment emits it at t_6. (b) Word-Error-Rate: two imperfect predictions with the same CER but different WER.

To exemplify the importance of prioritization, Table 1 shows that a CTC score is not a good proxy for some properties of the predicted alignment. It shows two different models with a similar training loss that have different WER and emission time, although trained on the same data.

Table 1: CTC score is not a good proxy for WER and latency of a system. The reference and the streaming models are CTC models with different architectures, which results in different predicted alignments. The latency of the streaming model is the delay in token emission time in comparison to the reference model. The Results are shown on the LibriSpeech test clean dataset.

| ASR Model Type | CTC Score | WER (%) | Latency (ms) |
| --- | --- | --- | --- |
| Reference | 0.0033 | 4.28 | 0 |
| Streaming | 0.003 | 5.43 | 217 |

To complement the CTC with an additional prioritization, we propose *Align With Purpose (AWP)* - **a Plug-and-Play framework** that allows enhancing a given property in the outputs of models trained with CTC while maintaining their transcription abilities. We add a loss term, $L_{AWP}$, that expresses a more subtle differentiation between alignments so that the final loss becomes $L = L_{CTC} + \alpha L_{AWP}$. Specifically, for a given property, we design a function $f_{prop}$ that receives an alignment as an input, and outputs an improved alignment with respect to the property. Then, we sample $N$ alignments based on the output probabilities of the pre-trained CTC model, apply $f_{prop}$ on the sampled alignments, to create $N$ pairs of alignments. Finally, we implement $L_{AWP}$ as hinge loss over the $N$ pairs, thus encouraging the model to increase the probability mass of the preferable alignments, as described in Figure 1.

Previous research has proposed controllability to model predictions. Liu et al. (2022) introduced an additional loss term that prioritizes a ranked list of alternative candidates during the training of generative summarization models. Specifically, for the case of hard alignment criteria like CTC, many proposed solutions are restricted to handling perfect alignments only, and some require intervention in the forward-backward algorithm (Tian et al., 2022; Yu et al., 2021; Shinohara & Watanabe, 2022; Yao et al., 2023; Laptev et al., 2023), as opposed to AWP. Alternative approaches address the imperfect alignments through additional loss terms, as seen in Prabhavalkar et al. (2018); Graves & Jaitly (2014). The aforementioned frameworks are less straightforward for implementation and might require a considerable amount of development time and optimization. In contrast, AWP offers a relatively simple implementation, requiring only a few lines of code.

To summarize, our main contributions are as follows: (1) Align With Purpose - a simple and general Plug-and-Play framework to enhance a general property in the outputs of a CTC model. (2) We show promising results in two properties that are independent of each other- we report an improvement of up to 590ms in latency optimization, and a relative improvement of 4.5% WER over the baseline models for the minimum WER (mWER) optimization. (3) We demonstrate the generality of our framework in terms of property selection, scale of the training dataset and architectural choice. To the best of our knowledge, these applications have never been demonstrated to work on a scale of data as large as ours. (4) The framework enables prioritization between both perfect and imperfect alignments.

We apply our approach to the ASR domain, specifically to models that are trained with CTC criterion. However, this method can be extended to other alignment-free objectives, as well as to other domains besides ASR.

## 2 CTC AND ALIGN WITH PURPOSE

The outline of this section is as follows: We start with a description of the CTC loss in subsection 2.1, followed by a detailed explanation of the proposed "Align With Purpose" method in subsection 2.2. Finally, we showcase two applications: low latency in subsection 2.3 and mWER in subsection 2.4.

## 2.1 CTC

The Connectionist Temporal Classification criterion (Graves et al., 2006) is a common choice for training seq2seq models. To relax the requirement of segmentation, an extra blank token $\emptyset$ that represents a null emission is added to the vocabulary $V$, so that $V' = V \cup \{\emptyset\}$.

Given a T length input sequence $\boldsymbol{x} = [x_1, ... x_T]$ (e.g. audio), the model outputs $T$ vectors $\boldsymbol{v}_t \in \mathbb{R}^{|V'|}$, each of which is normalized using the softmax function, where $\boldsymbol{v}_t^k$ can be interpreted as the probability of emitting the token $k$ at time $t$. An alignment $\boldsymbol{a}$ is a $T$ length sequence of tokens taken from $V'$, and $P(\boldsymbol{a}|\boldsymbol{x})$ is defined by the product of its elements:

$$P(\boldsymbol{a}|\boldsymbol{x}) = \prod_{t=1}^{T} p(\boldsymbol{a}_t|\boldsymbol{x}). \tag{1}$$

The probability of a given target sequence $\boldsymbol{y}$ (e.g. text) of length $U$, $\boldsymbol{y} = [y_1, ..., y_U]$ where $U \leq T$, is the sum over the alignments that yield $\boldsymbol{y}$:

$$P(\boldsymbol{y}|\boldsymbol{x}) = \sum_{\boldsymbol{a}:\boldsymbol{a}\in\mathcal{B}^{-1}(\boldsymbol{y})} p(\boldsymbol{a}|\boldsymbol{x}), \tag{2}$$

where $\mathcal{B}$ is the collapse operator that first removes repetition of tokens and then removes blank tokens.

The CTC objective function minimizes the negative log-likelihood of the alignments that yield $\boldsymbol{y}$, as seen in Eq. 3

$$L_{CTC}(\boldsymbol{x}) = -\log P(\boldsymbol{y}|\boldsymbol{x}). \tag{3}$$

By definition, the CTC criterion only enumerates over perfect alignments and weighs them equally. This means that CTC considers all perfect alignments as equally good Tian et al. (2022) and all imperfect alignments as equally bad Graves & Jaitly (2014)

## 2.2 ALIGN WITH PURPOSE

In this section, we present the suggested method, Align With Purpose (AWP). AWP complements the CTC loss with an additional loss term which aims at enhancing a desired property by adding a more subtle prioritization between alignments.

Given a desired property to enhance, we define a property-specific function $f_{prop} : V'^T \rightarrow V'^T$, that takes as input an alignment $\boldsymbol{a}$ and returns an alignment $\bar{\boldsymbol{a}}$ with the same length. $f_{prop}$ is designed to output a better alignment w.r.t. the property. During training, at each step we sample $N$ random alignments according to the distribution induced by the output of the seq2seq model, such that $\boldsymbol{a}_t^i \sim \boldsymbol{v}_t$ for $t \in [1..T]$ and $i \in [1..N]$ (see Appendix B for more details on the sampling method). We then apply $\bar{\boldsymbol{a}}^i = f_{prop}(\boldsymbol{a}^i)$ to obtain better alignments. This results in $N$ pairs of alignments $(\boldsymbol{a}^i, \bar{\boldsymbol{a}}^i)$, where $\bar{\boldsymbol{a}}^i$ is superior to $\boldsymbol{a}^i$ in terms of the property. Finally, to enhance the desired property the model is encouraged to increase the probability mass of $\bar{\boldsymbol{a}}^i$, by applying hinge loss on the probabilities of the alignment pairs:

$$L_{AWP}(x) = \frac{1}{N} \sum_{i=1}^{N} max\{P(\boldsymbol{a}^i|x) - P(\bar{\boldsymbol{a}}^i|x) + \lambda, 0\}, \tag{4}$$

where $\lambda$ is a margin determined on a validation set. See Fig. 1 for an illustration of the proposed framework.

As pointed out in (Graves & Jaitly, 2014; Prabhavalkar et al., 2018), sampling from a randomly initialized model is less effective since the outputs are completely random. Therefore, we train the model to some extent with a CTC loss as in Eq. 3, and proceed training with the proposed method.

Putting it all together, the training loss then becomes:

$$L(x) = L_{CTC}(x) + \alpha L_{AWP}(x), \tag{5}$$

where $\alpha$ is a tunable hyper-parameter that controls the trade-off between the desired property and the CTC loss.

## 2.3 APPLICATIONS: LOW LATENCY

Streaming ASR systems with low latency is an active research field, as it serves as a key component in many real world applications such as personal assistants, smart homes, real-time transcription of meetings, etc. (Song et al., 2023). To measure the overall latency of a system, three elements should be considered: data collection latency (**DCL**) which is the future context of the model, computational latency (**CL**) and drift latency (**DL**), as defined by Tian et al. (2022). For the latter, we slightly modified their definition, see Appendix A.2 for more details. We also leave the CL component out of the scope of this work as it is sensitive to architectural choice, hardware, and implementation. Thus, we denote by **TL=DCL+DL** the total latency of the system.

Several techniques were suggested to reduce the TL: input manipulation (Song et al., 2023), loss modification (Tian et al., 2022), loss regularization (Yu et al., 2021; Yao et al., 2023; Shinohara & Watanabe, 2022; Tian et al., 2023), and architectural choice (Pratap et al., 2020). These methods are specific to low latency settings, or require intervention in the forward-backward algorithm. See Appendix D for more details and comparison to other works.

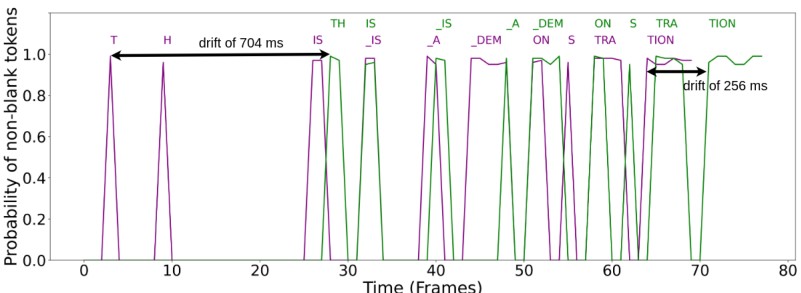

Figure 3: Drift in emission time in a CTC model. Bottom purple text: An offline Stacked ResNet model with symmetric padding, with 6.4 seconds of context divided equally between past and future contexts. Top green text: An online Stacked Resnet with asymmetric padding, with 430ms future context and 5.97 seconds past context. It can be seen that the output of the online model has a drift ≥200 ms.

One way to reduce the DCL is by limiting the future context of the model. In attention based models it can be achieved by left context attention layers (Yu et al., 2021), and in convolutional NN it can be achieved using asymmetrical padding (Pratap et al., 2020). However, Pratap et al. (2020) have shown that training with limited future context results in a drift (delay) in the emission time of tokens (DL), as can be seen in Fig. 3. The cause of the drift was explained by Wang et al. (2020), who made the observation that less future context deteriorates performance. Therefore, by delaying the emission time, the model effectively gains more context, which in turn improves its performance.

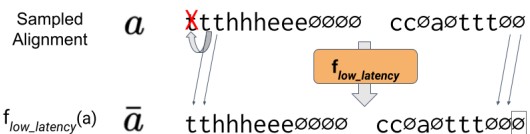

Figure 4: Defining $f_{low\_latency}$. To obtain $\bar{a}$, we shift the sampled alignment $a$ one token to the left, starting from a random position (second token in this example) within the alignment, and pad $\bar{a}$ with a trailing blank token, marked by a black rectangle

To mitigate the DL using AWP, given an alignment $a$, we sample a random position within it, and shift $a$ one token to the left from that position to obtain $\bar{a}$ as seen in Fig. 4. Clearly, tokens emission time of $\bar{a}$ is one time step faster than $a$ starting from the random position. By limiting the initial shift position to correspond to tokens that are repetitions, we ensure that the collapsed text of $\bar{a}$ remains the same as $a$. To make $\bar{a}$ a $T$ length alignment, we pad it with a trailing blank token.

Formally, we define the function $f_{low\_latency}$. Given an alignment $a$, define the subset of indices $[j_1, .., j_{T'}] \subseteq [2..T]$ as all the indices such that $a_{j_k} = a_{j_k - 1}$, meaning that $a_{j_k}$ is a repetition of the

previous token. Then we sample a random position $j$ from $[j_1, .., j_{T'}]$, and obtain $\bar{\boldsymbol{a}}$:

$$\bar{\boldsymbol{a}}_t = \begin{cases} \boldsymbol{a}_t & \text{if } t < j - 1 \\ \boldsymbol{a}_{t+1} & \text{if } j - 1 \leq t < T \\ \emptyset & \text{if } t == T \end{cases} \tag{6}$$

## 2.4 APPLICATIONS: MINIMUM WORD ERROR RATE

The most common metric to assess an ASR system is the word error rate (WER). Nevertheless, training objectives such as CTC do not fully align with this metric, resulting in a gap between the training and testing criteria. Therefore, the system's performance could improve by adding a prioritization over the imperfect alignments w.r.t. their WER. This gap was previously addressed by Graves & Jaitly (2014), who suggested approaching it by minimizing the expected WER but this method requires extensive optimization. Prabhavalkar et al. (2018) suggested a similar objective for training attention models with the cross-entropy (CE) loss.

Figure 5: Defining $f_{mWER}$. Given a target transcription 'the cat', the (upper) sampled alignment yields the text 'tha cet', which has 100% WER. Substituting the occurrences of the token 'e' with the token 'a' produces the text 'tha cat', which has 50% WER.

As illustrated in Figure 5, to apply AWP for mWER training, we define $f_{mWER}$. Given a sampled **imperfect** alignment $\boldsymbol{a}$ and a GT transcription $\boldsymbol{y}$, to obtain $\bar{\boldsymbol{a}}$ we select the word in the collapsed text $\mathcal{B}(\boldsymbol{a})$ which requires the minimum number of substitutions in order for it to be correct. Then we fix the alignment of this word according to the GT, so that the number of word errors in $\mathcal{B}(\bar{\boldsymbol{a}})$ is reduced by 1.

## 3 EXPERIMENTAL SETUP

We evaluate AWP on two end-tasks: low latency and mWER, by conducting experiments using multiple architectures and different scales of datasets. The general settings are listed below. For more details on the setup see Appendix A.

**Datasets**. We examine our framework on 3 scales of the data, ranging from 1K to 280K hours. The small scale dataset is LibriSpeech (Panayotov et al., 2015) (LS-960). The medium scale dataset consists of 35K hours curated from LibriVox[2] (LV-35K). The large scale is an internal dataset of 280K hours of audio-transcript pairs (Internal-280K), which, to the best of our knowledge, is the largest dataset that was used to train a low-latency model. We test our framework on the test splits of LibriSpeech.

**Architecture.** To validate that our method is invariant to architecture selection, we trained 3 different architectures: Stacked ResNet (He et al., 2016), Wav2Vec2 (Baevski et al., 2020) and a Conformer (Gulati et al., 2020) model.

We used a pre-trained Wav2Vec2 base model with 90M parameters, available on HuggingFace [3]. This architecture was used in the mWER training experiments.

The Conformer employed is a medium-sized model with 30.7M parameters. As an offline model, given its attention layers, its future context was not limited. To transition from an offline to an online model, during inference, the right context (DCL) was restricted to 430ms. The left context was also limited to 5.57s, resulting in a 6s of context in total. The model consumes the input in chunks, similarly to Tian et al. (2022).

---

[2] http://www.openslr.org/94/
[3] https://huggingface.co/facebook/wav2vec2-base-100k-voxpopuli

Lastly, the Stacked ResNet consists of 66M parameters. This architecture can be implemented in a streaming manner and can be highly optimized for edge devices. Therefore, it's a good fit for an online system in a low-resource environment. In our implementation, the model has 6.4s of context. In the offline version of this model, the 6.4s are divided equally between past and future context, i.e. it has a DCL of 3.2s. The online version, implemented with asymmetric padding as suggested by Pratap et al. (2020), has also 6.4s context, but its DCL is only 430ms, which makes it feasible to deploy it in online ASR systems. We used the offline implementation in both end tasks- as a baseline in the mWER training and as an offline model in the low latency training.

**Decoding**. Models were decoded using an in-house implementation of a beam search decoder described in (Graves & Jaitly, 2014).

**Training**. To test the effectiveness of AWP, we train the models for several epochs, and then apply our framework, namely adding the AWP loss to the CTC loss as stated in Eq. 5. The epoch in which we start to apply our framework is denoted as 'start epoch' in tables 2, 3.

The Wav2Vec2 model was finetuned using SpecAugment (Park et al., 2019) on LS-960 using the Adam optimizer (Kingma & Ba, 2014) with a flat learning rate (LR) scheduler Baevski et al. (2020). The Conformer was trained on LS-960 with the same training scheme described in Gulati et al. (2020). The Stacked ResNet models were optimized with RAdam optimizer (Liu et al., 2019) with a ReduceLROnPlateau scheduler[4] on different data scales.

**Evaluation Metrics**. To test AWP, in the mWER setting we evaluated models with a standard implementation of WER. In the low latency setting we evaluated DL and TL as defined in section 2.3.

## 4  RESULTS

In this section, we present the results achieved by training using AWP in the low latency and mWER applications.

### 4.1  LOW LATENCY

Table 2 shows the results when training the Stacked ResNet model on small, medium and large scales of data. We can see a clear trend across all scales that the AWP training successfully decreases the DL. The DL for each model is computed in comparison to its relevant offline model. It can also be seen that some models achieve negative DL, meaning that the TL is reduced beyond its expected lower bound induced by the DCL. In most cases, achieving such low TL solely by reducing the architectural future context using another padding optimization would not have been possible. This trend also holds for various implementations of the property function, as can be seen in Appendix C.

Table 2 also shows our implementation (or adjustment of public code) of selected prior work in the field of low latency in CTC training: BayesRisk CTC (Tian et al., 2022), Peak First CTC (Tian et al., 2023) and TrimTail (Song et al., 2023). It can be seen that AWP outperforms the other methods, both in terms of WER and latency. See Appendix D and F for more details.

We can also see that as the scale of the data increases, the WER decreases. This statement holds independently for the offline models and for the online models, and remains valid also after adding the AWP loss. This shows that AWP does not affect the ability of the model to improve its basic transcription capabilities using larger scales of data, which aligns with previous observations on large scale training Baevski et al., 2020; Radford et al., 2023.

In almost all the experiments, the WER increases with the latency reduction. This is a known trade-off between latency and accuracy as reported in prior work (Pratap et al., 2020). The choice of the operating point in terms of the balance between latency and accuracy can be determined by the weight of the AWP loss, $\alpha$, and the scheduling of when we add the AWP loss ('start epoch'), as can be seen in Fig. 6.

The Conformer with AWP experiment demonstrates a DL reduction and a trade-off between latency and accuracy, thus affirming that AWP is not limited to a specific architecture. Given the unrestricted

---

[4]`https://pytorch.org/docs/stable/generated/torch.optim.lr_scheduler.ReduceLROnPlateau.html`

future context of the offline model, DCL matches the input size, making TL measurement irrelevant. The online model, not trained in an online fashion (see Appendix A.2) is expected to lack a DL. Yet, AWP can reduce the DL to negative values, which in turn reduces the TL.

Table 2: Low Latency model training with and without AWP on different data scales, and with other frameworks. 'Start Epoch' denotes the step that we added AWP, and was chosen based on a list of milestones WER of the online model. The different entries in the table are reported based on the best checkpoint in terms of WER, for each model separately. Results are on Libri Test-Clean.

| Model | Training Data | Start Epoch | DL (ms) | TL (ms) | WER |
|---|---|---|---|---|---|
| Stacked ResNet Offline | Internal-280K | - | 0 | 3.2K | 2.34 |
| Stacked ResNet Online | Internal-280K | - | 249 | 679 | 2.6 |
| +AWP | Internal-280K | 5.7 | 50 | 480 | 2.71 |
| Stacked ResNet Offline | LV-35K | - | 0 | 3.2K | 2.42 |
| Stacked ResNet Online | LV-35K | - | 341 | 771 | 2.72 |
| +AWP | LV-35K | 0.1 | -251 | 179 | 3.28 |
| Stacked ResNet Offline | LS-960 | - | 0 | 3.2K | 3.72 |
| Stacked ResNet Online | LS-960 | - | 278 | 708 | 4.06 |
| +AWP | LS-960 | 0.9 | -79 | 351 | 4.38 |
| +Peak First CTC | LS-960 | - | 186 | 616 | 4.41 |
| +TrimTail | LS-960 | - | -76 | 354 | 4.46 |
| +Bayes Risk | LS-960 | - | 63 | 493 | 4.78 |
| Conformer Offline | LS-960 | - | 0 | - | 3.7 |
| Conformer Online | LS-960 | - | 2 | 432 | 3.75 |
| +AWP | LS-960 | 12 | -172 | 263 | 3.74 |

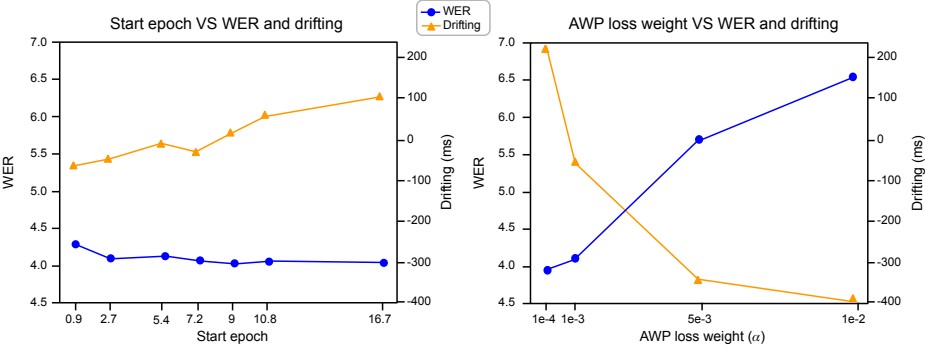

Figure 6: The effect of the 'start epoch' and $\alpha$ on the word error rate (WER) and the drifting (DL). While the 'start epoch' has more effect on the DL, the WER only slightly changes (Left). To test the effect of the $\alpha$, we fixed the 'start epoch' to 2.7 and applied AWP with different weights. The selection of $\alpha$ has a significant impact on both the latency and the WER (Right).

## 4.2 MINIMUM WORD ERROR RATE

Table 3 shows a significant relative improvement of 4-4.5% in Word Error Rate (WER) when applying AWP. It also shows that AWP yields similar results to our adaptation of MWER optimization (MWER_OPT), as suggested by Prabhavalkar et al. (2018) (originally designed for soft alignments models trained with a CE loss). This emphasizes that AWP is competitive with application-specific methods while maintaining its general nature and simplicity. Improvement in WER also gained with various implementations of the property function, as can be seen in Appendix C.

Furthermore, our proposed framework proves to be versatile, as it successfully operates on both streaming (Stacked ResNet) and offline (Wav2Vec2) architectures. The ability of our approach to adapt to different architectures highlights its applicability across various ASR systems.

Table 3: WER of baseline models and of models optimized for WER with AWP or MWER_OPT.

| Model | Start Epoch | % WER Libri Test-Clean (% Relative improvement) | % WER Libri Test-Other (% Relative improvement) |
|---|---|---|---|
| Stacked ResNet | - | 2.63 | 7.46 |
| +AWP | 4.3 | 2.57 (2.2) | 7.16 (4) |
| +MWER_OPT | 4.3 | 2.54 (3.4) | 7.31 (2) |
| Wav2Vec | - | 2.38 | 5.82 |
| +AWP | 2.3 | 2.33 (2.1) | 5.56 (4.5) |

## 5 DISCUSSION & FUTURE WORK

The results obtained from our study provide valuable insights regarding the potential for improvement in ASR models trained with the CTC criterion. Although not tested, this framework could be easily applied to other hard-alignment criteria such as Transducer (Graves, 2012). Furthermore, by adapting and extending the concepts from our framework, it may be possible to enhance soft-alignment methods, even in domains beyond ASR.

In addition, an intriguing aspect for future research is the formalization of the properties that can be enhanced using AWP. By establishing a formal framework, researchers can systematically identify, define, and prioritize the properties to be enhanced. This can lead to targeted improvements and a deeper understanding of the impact of different properties on ASR performance. Finally, our study showcases the capability of enhancing a single property at a time. In some applications, multiple properties should be enhanced simultaneously, potentially leading to better performance. It could be especially intriguing in scenarios where the distinct properties exhibit a trade-off, like the low latency and WER properties. Utilizing AWP on both properties can provide a more nuanced control over their trade-off.

## 6 LIMITATION & BROADER IMPACT

Although the AWP framework is relatively easy to use, its main limitation is that one needs to think carefully about the property function $f_{prop}$. When formulated elegantly, the implementation is straight forward.

The proposed AWP framework enables one to enhance a desired property of an ASR model trained with CTC. As mentioned in 5, this method can be applied or adapted to domains other than ASR. On the choice of the property to enhance, especially in generative AI, one should be thoughtful not to increase bias, malicious or racist content of models.

## 7 CONCLUSIONS

The dichotomy between perfect and imperfect alignments in CTC highlights its limitation in capturing additional alignment properties, which is a key requirement in many real-world applications. To overcome this limitation, we introduce Align With Purpose, a general Plug-and-Play framework designed to enhance specific properties in models trained using the CTC criterion. Our experimental results demonstrate promising outcomes in two key aspects: latency and minimum Word Error Rate optimization. Importantly, these optimizations are independent of each other, highlighting the versatility of AWP. The reduced latency achieved by our approach indicates faster transcription while maintaining transcription quality even with significantly reduced drift. Furthermore, our improved WER emphasizes the importance in enabling differentiation between imperfect alignments for enhancing the transcription quality of ASR systems. One of the strengths of AWP lies in its generality. It offers flexibility in selecting specific alignment properties, applies to large-scale training datasets, and is versatile to architectural choice. Our method does not require modifications to the CTC loss function and can be implemented using only a few lines of code.

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

## A    APPENDIX: ADDITIONAL EXPERIMENTAL SETTINGS

### A.1    GENERAL EXPERIMENTAL SETTINGS

**Datasets**. For the small scale, we train models on the LibriSpeech dataset (Panayotov et al., 2015), which consists of 960 training hours (LS-960). For the medium scale, we train models on a 35K hours curated subset of LibriVox[5] (LV-35K), where samples with low confidence of a reference model were filtered out. For the large scale, we train models on an internal dataset of 280K hours of audio-transcript pairs (Internal-280K), which, to the best of our knowledge, is the largest dataset that was used to train a low-latency model. We test our framework on the test splits of LibriSpeech. Audio is sampled at 16KHz, 16 bits/sample.

**Architecture.** We trained Stacked ResNet (He et al., 2016), Wav2Vec2 (Baevski et al., 2020), and Conformer (Gulati et al., 2020) models. The ResNet consists of 20 ResNet blocks (66M parameters). For the Wav2Vec2, we used a pre-trained version of the base Wav2Vec2 model (90M parameters) available on HuggingFace [6]. The model was pre-trained for 30 epochs on the 100K hours from VoxPopuli dataset (Wang et al., 2021). Lastly, we trained a medium Conformer model (30.7M parameters) implemented according to the setup described in (Gulati et al., 2020). All models output 29 English lower-case characters, including apostrophes, spaces, and blank tokens.

Regarding the Stacked ResNet model, we extracted 80-channel Mel filter-banks features computed from a 32ms window with a stride of 16ms. For each frame, we stacked the filter banks with a first and second derivative, resulting in a 240-dimensional input vector. We down-sample the audio input from 16ms to 32ms by applying MaxPool layer within the first layer of the first Stacked ResNet block, then stacked 20 ResNet blocks (He et al., 2016) with a kernel size of 5. Skip connections are added every 4 ResNet blocks. The model consists of 66M parameters in total. This architecture induces 6.4 seconds of context in total. The results are shown using an exponential moving average (EMA) model, which is aggregated alongside the model.

**Decoding**. Models were decoded using an in-house implementation of a beam search decoder described in (Graves & Jaitly, 2014), using a beam size of 100, and two language models: an open-source 5-gram language model[7] (WordLM) trained on the LibriSpeech LM corpus, and a character-level language model (CharLM) that we trained on the same corpus. The beam search picks transcriptions $y$ which maximize the quantity $L(y)$ defined by:

$$L(y) = P_{acoustic}(y|x) + \beta P_{CharLM}(y) + \gamma P_{WordLM}(y) \tag{7}$$

where $\beta = 0.8$ and $\gamma = 0.8$ are the CharLM and WordLM weights, respectively.

**Text Normalization**. We used an in-house implementation of text normalization to remain in a vocabulary of 29 English characters.

### A.2    LOW LATENCY EXPERIMENTAL SETTINGS

**Architecture**. Experiments detailed in this section are conducted with the Stacked ResNet and Conformer architectures described in A.1.

---

[5]http://www.openslr.org/94/

[6]https://huggingface.co/facebook/wav2vec2-base-100k-voxpopuli

[7]https://www.openslr.org/11/

The ResNet architecture can be implemented in a streaming manner and can be highly optimized for edge devices. Therefore, it's a natural choice for an online system in a low- resource environment. Our offline version of it has 6.4 seconds of context in total, divided equally between past and future contexts. Although the model can be implemented in a streaming fashion, it has a large DCL of 3.2s. The online version has a similar architecture and the same total context, but it has a DCL of 430ms, achieved by asymmetric padding as suggested by Pratap et al. (2020). The small DCL of this model makes it feasible to deploy it in an online ASR system.

For the Conformer, we only trained it in an offline fashion, with or without AWP. To transform the Conformer into an online model, at inference the DCL was to restricted 430ms and the left context was restricted to 5.57s. The input was fed into the model chunk by chunk, similarly to Tian et al. (2022).

**Training**. For training the ResNet with AWP on LS-960, LV-35K, and Internal-280K, the hyper-parameters $\alpha$ and $\lambda$ were set to 0.001, 0.001, 0.0005, and 0.01, 0, 0, respectively.

RAdam optimizer (Liu et al., 2019) with $\alpha = 0.9$, $\beta = 0.999$ and weight decay of 0.0001 were used. We set the LR to 0.001, with a ReduceLROnPlateau scheduler [8].

To train the Conformer with AWP, a scheduling for $\alpha$ was required - 0.1 at commencement and 1e-6 after 7K training steps. $\lambda$ remained constant and was set to 0.01.

For both models, we set $N = 5$, the number of sampled alignments.

**Measuring DL**. Measuring the DL of an online model is relative to the offline model of the same architecture that was trained on the same data. To measure the DL, we force-align the target transcript (GT) of the offline and online models independently and take the difference between the index of the first appearance of each token in the two force-aligned texts. Then, we take the average difference between all tokens. We empirically verified that the DL of the offline models compared to the true emission time is negligible, and for some instances, it even appears to be negative. This behavior was also observed by (Tian et al., 2023).

### A.3 MINIMUM WORD ERROR RATE EXPERIMENTAL SETTINGS

**Architecture**. In this setting, we applied AWP to a Stacked ResNet and a Wav2Vec2 models, as described in subsection A.1. The Stacked ResNet model that was used here is the same as the offline model described in subsection A.2.

**Training**. The baseline Stacked ResNet model was pre-trained on the Internal-280K dataset. Then we continue its training solely on LS-960 for 4.3 epochs before we apply AWP. The AWP hyper-parameters were $\alpha = 0.1$, $\lambda = 0$. The baseline and the model with AWP were trained for 4.2 additional epochs, reaching 8.5 epochs in total. We used the RAdam optimizer (Liu et al., 2019) with the same hyper parameters as in subsection A.1.

The Wav2Vec2 baseline model was finetuned with SpecAugment (Park et al., 2019) (with p=0.05 for time masking and p=0.0016 for channel masking) solely on LS-960 for 2.3 epochs before we applied AWP, and both the baseline and the AWP models were trained for another 27.5 epochs. We used the Adam optimizer (Kingma & Ba, 2014) for this training, as well as a flat LR scheduler Baevski et al. (2020). AWP hyper-parameters were set to $\alpha = 0.05$ and $\lambda = 0$.

While training all models with AWP, we used a softmax temperature of 0.5 for the sampling of the $N$ alignments. Additionally, we set $N = 10$ under these settings.

## B SAMPLING METHOD

Throughout our experiments, we used the standard torch library for sampling[9]. To verify that the results weren't compromised by the lack of differentiability of the sampling process, we conducted similar experiments with Gumbel Softmax (Jang et al., 2016). As can be seen in Fig. 7, the Gumbel Softmax had not effect on results.

---

[8] https://pytorch.org/docs/stable/generated/torch.optim.lr_scheduler.ReduceLROnPlateau.html

[9] https://pytorch.org/docs/stable/data.html#torch.utils.data.Sampler

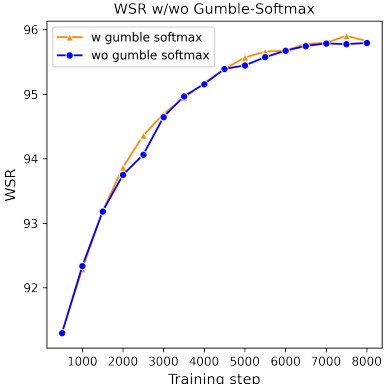 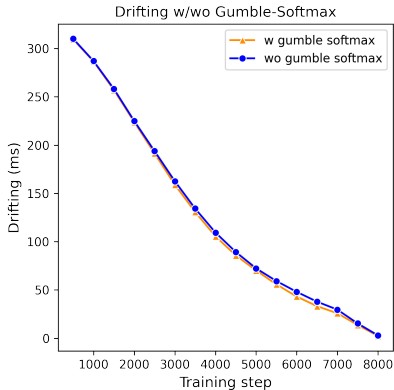

Figure 7: A Stacked ResNet low latency model, with AWP applied after 2.7 epochs. The sampling method was either the standard torch implementation or Gumbel Softmax. It can be seen that the sampling method has minimal impact on the drifting (DL) and on the WSR, which is the word success rate (100-WER).

## C  VARIANTS OF THE PROPERTY FUNCTION

For both applications, low latency and WER, we verified that AWP is robust to the choice of the property function. Table 4 shows different implementations for the low latency function presented in section 2.3. Specifically, instead of shifting alignments by one token to the left, we shift them by multiple tokens in random positions. It can be seen that gradual changes in alignments, i.e. fewer shifts, have a positive effect on the latency. Although the changes are gradual, the overall improvement is far greater than a shift in a few tokens, as the model improves throughout the training process, and at every training step it is optimized to improve its (current) latency by a shift of a few tokens.

Table 4: Different property function implementations for the Low Latency application. The different implementations vary in the number of tokens to shift and are reported after the same number of training steps.

| Tokens Shifted | DL (ms) | WER |
|---|---|---|
| 1 | -79 | 4.38 |
| 2 | -68 | 4.46 |
| 4 | 21 | 4.39 |

As for the WER application, table 5 shows a variation for the property function defined in section 2.4, when correcting 2 words instead of 1.

Table 5: Different property function implementations for the WER application. The different implementations vary in the number of words to correct and are reported after the same number of training steps.

| Words Corrected | % WER Libri Test-Clean | % WER Libri Test-Other |
|---|---|---|
| 1 | 2.57 | 7.16 |
| 2 | 2.54 | 7.29 |

## D    DETAILS AND RESULTS OF PRIOR WORK

In this section, we present both the findings and the specific details of prior work that address latency reduction in CTC training. Since most of the prior work used different datasets and architectures, we implemented or adjusted existing implementations to work in our setting.

**Methods**. The different methods we tried are TrimTail (Song et al., 2023), Peak First CTC (PFR) (Tian et al., 2023) and Bayes Risk CTC (BRCTC) (Tian et al., 2022).

**Implementation**. We implemented TrimTail and PFR in accordance with the relevant paper. In the case of BRCTC, we utilized an open-source implementation from ESPnet library[10].

**Experimental Settings And Results**. We trained the models on LS-960, using the same online ResNet model as described in 3. Decoding, training optimizer and scheduler are the same as in A.2

As for the hyper-parameters, the TrimTail has a max trim value, PFR has a weight and BRCTC has a risk factor value. These parameters are responsible for controlling the method's effectiveness. For the PFR we used a dynamic weight $w$ with a fixed ratio $\alpha$, s.t. $\frac{loss\_CTC}{\alpha * loss\_PFR} = w$.

We conducted extensive research in order to find the best hyper-parameters that fit our model and setup to achieve the best drift latency with minimum degradation in the WER results. The best parameters we found are max_trim_value=50 for TrimTail, $\alpha$=0.001 for PFR, and risk_factor=200 for Bayes Risk. We used this set of parameters for the results reported in table 2.

Table 6 presents the full results of these models using various ranges of hyper-parameters.

Table 6: Other low latency methods results. DL was calculated using the Offline model trained on LS-960 dataset. Results are on Libri Test-Clean.

| Model | Hyper-parameter | DL (ms) | TL (ms) | WER |
|---|---|---|---|---|
| Peak First CTC | $\alpha$=0.01 | 198 | 628 | 4.22 |
| Peak First CTC | $\alpha$=0.001 | 235 | 665 | 4.11 |
| Peak First CTC | $\alpha$=0.0001 | 186 | 616 | 4.41 |
| TrimTail | max_trim=30 | 47 | 477 | 4.38 |
| TrimTail | max_trim=50 | -76 | 354 | 4.46 |
| TrimTail | max_trim=70 | -126 | 304 | 4.79 |
| Bayes Risk | risk_factor=100 | 199 | 629 | 4.21 |
| Bayes Risk | risk_factor=200 | 63 | 493 | 4.78 |
| Bayes Risk | risk_factor=250 | 91 | 521 | 4.82 |

## E    FURTHER COMPARISON BETWEEN AWP AND PRIOR WORK

To enable additional comparison between AWP and prior work, table 7 shows that for a similar WER value, the latency varies across methods. This specific WER value was taken since this was the minimal WER that one of the other methods achieved. It can be seen that AWP outperforms in terms of latency.

---

[10]https://github.com/espnet/espnet

Table 7: AWP and other low latency methods results, when the WER is similar across the different methods. The models were trained on LS-960 dataset and are the same models used in table 2. DL was calculated using the Stacked ResNet Offline model trained on LS-960 dataset. Results are on Libri Test-Clean.

| Method | WER | DL (ms) |
|--------|-----|---------|
| Peak First CTC | 4.7 | 186 |
| Bayes Risk | 4.78 | 64 |
| TrimTail | 4.81 | -59 |
| AWP | 4.75 | -96 |

## F DETAILED LOW LATENCY EXPERIMENTAL RESULTS

Table 8: Full results with and without AWP, with other frameworks and on different data scales. Results are on Libri Test-Clean.

| Model | Training Data | Start Epoch | DL (ms) | TL (ms) | WER |
|-------|---------------|-------------|---------|---------|-----|
| Stacked ResNet Offline | Internal-280K | - | 0 | 3.2K | 2.34 |
| Stacked ResNet Online | Internal-280K | - | 249 | 679 | 2.6 |
| +AWP | Internal-280K | 0.03 | 33 | 463 | 3.13 |
| +AWP | Internal-280K | 5.7 | 50 | 480 | 2.71 |
| Stacked ResNet Offline | LV-35K | - | 0 | 3.2K | 2.42 |
| Stacked ResNet Online | LV-35K | - | 341 | 771 | 2.72 |
| +AWP | LV-35K | 0.1 | -251 | 179 | 3.28 |
| Stacked ResNet Offline | LS-960 | - | 0 | 3.2K | 3.72 |
| Stacked ResNet Online | LS-960 | - | 278 | 708 | 4.06 |
| +AWP | LS-960 | 0.9 | -79 | 351 | 4.38 |
| +AWP | LS-960 | 2.7 | -54 | 376 | 4.11 |
| +AWP | LS-960 | 5.4 | -18 | 412 | 4.13 |
| +AWP | LS-960 | 7.2 | -32 | 398 | 4.07 |
| +AWP | LS-960 | 9 | -24 | 406 | 3.92 |
| +AWP | LS-960 | 10.8 | 53 | 483 | 4.06 |
| +AWP | LS-960 | 16.7 | 136 | 566 | 3.92 |
| +Peak First CTC | LS-960 | - | 186 | 616 | 4.41 |
| +TrimTail | LS-960 | - | -76 | 354 | 4.46 |
| +Bayes Risk | LS-960 | - | 63 | 493 | 4.78 |
| Conformer Offline | LS-960 | - | 0 | - | 3.7 |
| +AWP | LS-960 | 12 | -172 | - | 3.74 |

Table 9: Low Latency model training w/ & w/o AWP, with different AWP loss weight. Results are on Libri Test-Clean

| Model | Training Data | AWP loss weight | DL (ms) | TL (ms) | WER |
|-------|---------------|-----------------|---------|---------|-----|
| Stacked ResNet Offline | LS-960 | - | 0 | 3.2K | 3.72 |
| Stacked ResNet Online | LS-960 | - | 278 | 708 | 4.06 |
| +AWP | LS-960 | 0.01 | -402 | 28 | 6.52 |
| +AWP | LS-960 | 0.005 | -346 | 84 | 5.69 |
| +AWP | LS-960 | 0.001 | -54 | 376 | 4.11 |
| +AWP | LS-960 | 0.0005 | 114 | 566 | 3.9 |

