# OpenReview forum: "Align With Purpose: Optimize Desired Properties in CTC Models with a General Plug-and-Play Framework"
_ICLR.cc/2024/Conference — ICLR 2024 poster_

### Official Review · Reviewer_4pZf · 2023-10-21

**Soundness:** 3 good
**Presentation:** 4 excellent
**Contribution:** 3 good
**Rating:** 8
**Confidence:** 4

**Summary:**

This paper proposes a novel method for improving the desired properties (latency and accuracy) of CTC-based speech recognition models. The core idea of the proposed method, called AWP, is to distinguish different alignment paths by prioritizing the one exhibiting the better property using an additional loss term. To promote such properties, simple rule-based strategies are employed to modify the alignment. For example, one may shift 1 token to generate a ‘worse’ (delayed) alignment path. Experiments conducted on online/offline ASR models show that AWP can boost the desired properties compared to previous baselines.

**Strengths:**

* The paper is very well-written and easy to understand. Especially, the introduction and related work sections are such a joy to read.
* The comparative experiments against previous methods (including very recent ones) are conducted under the same condition. The result demonstrates the effectiveness of AWP (Table 2).
* The proposed method appears to be a novel CTC modification utilizing a sampling-based hinge loss function. The approach is clearly different from previous methods pursuing similar objectives.

**Weaknesses:**

* In Table 2, only the ‘Stacked ResNet Online’ model case is compared with other methods. It would strengthen the claim if there exists more comparison for Conformer-Online + AWP, Peak First CTC, Trim-Tail, etc.
* While the latency reduction part presents extensive experimental results (various training data sizes, comparison, ablation, ...), there are not many results on minimum WER training. Furthermore, the gain from minimum WER training is marginal.
* There seems to be room for improvement; for example, how about increasing the number of shifted frames (tokens) instead of selecting just one? How about applying AWP together with Trim-Tail? I appreciate the simplicity of the proposed method, but I am also curious about the limitations of this method.

**Questions:**

* What is WSR in Figures 6 and 7? Is it (100 – WER)?
* It seems that AWP needs random sampling at each training step. How many alignments (N) do you sample for each step? How does AWP affect the overall training time/resource usage?

---

> ### Author Response · Authors · 2023-11-22
> **Response to Reviewer 4pZf**
>
> We thank the reviewer for the valuable feedback, which assisted us clarify some aspects of the work and update the text accordingly. We value the acknowledgement of the strengths of our work, in particularly its thorough comparison to previous work and its generality. In the following, we did our best to address every concern or question raised by the reviewer.
>
> **Strengthening table 2 by adding more experiments with an online Conformer**
>
> Table 2 presents an extensive research, to allow the different methods to work under the same settings as AWP, and we thank the reviewer for acknowledging it as a strength of the paper. The wide array of experiments conducted with a Conformer weren’t fully matured by the ICLR deadline. However, we enhanced table 2 by adding an online Conformer with AWP and updated the text accordingly. We kindly refer the reviewer for the ‘common response’ part for the details.
>
> **While the latency reduction part presents extensive experimental results, there are fewer results on minimum WER training, and their gain is marginal.**
>
> The evidence for the successful application of AWP to the low latency application is stronger and wider. Yet, although more modest, the WER part presents 2 model architectures, and another method for optimizing mWER was added to table 3 for additional reference (see the ‘common response’ part). The gain in this part is moderate compared to prior work.  Graves & Jaitly, 2014 reported a relative improvement ranging from 3.7% to 9% on the Wall Street Journal dataset. Prabhavalkar et al,. 2018 reported an improvement ranging between 4.1% to 7.4%, and [1] reported a 5% improvement, both on non-public datasets. We believe that although the gains for the latency application are more prominent than the WER application, the relative improvement in the WER application demonstrates the versatility and the generality of AWP.
>
> **Can results be improved by increasing the number of shifted frames (tokens), or by applying AWP together with Trim-Tail, for example?**
>
> As demonstrated in table 2, AWP achieves better results than the other methods both in latency and in accuracy, including reaching a negative DL. Increasing the number of shifted frames yielded a similar trend, but our empirical findings suggest that gradual changes were favored. We added appendix C to discuss the variations of the property function, we kindly refer the reviewer to the ‘common response’.
> Exploring the combination of AWP with other methods like Trim-Tail can be very promising. However, it can be expected that the trade off between latency and accuracy may still play an important role, as decreasing the latency even further is expected to increase the WER.
>
> **WSR is figured 6 and 7**
>
> WSR is the word success rate, 100-WER. We modified figure 6 to present WER, and changed the caption of figure 7.
>
> **How many alignments (N) do you sample for each step? How does AWP affect the overall training time/resource usage?**
>
> We thank the reviewer for the question, we added this information to appendix A. Under the low latency setting, we set N=5 and for the WER setting we set N=10, based on empirical observations.
> As mentioned, sampling occurs every training step, and training with AWP is, empirically, 2-3 times slower than training a baseline model.
>
> [1]  M. Shannon, “Optimizing expected word error rate via sampling for speech recognition”

---

### Official Review · Reviewer_jMFR · 2023-10-30

**Soundness:** 4 excellent
**Presentation:** 4 excellent
**Contribution:** 4 excellent
**Rating:** 8
**Confidence:** 5

**Summary:**

The paper proposes a plug-and-play framework to CTC loss so as to improve the performance of trained model on a specific perspective. The preference is achieved by a hinge loss calculated between an example and a better example. The experiments on ASR with different data scale show that the proposed method can help model to recognize text promptly or accurately.

**Strengths:**

The design of Align With Purpose is nice and interesting. The method makes use of a fact that in conventional CTC loss, all the perfect alignment are treated equally. In this case, the preference can be achieved by helping model compare to possible paths. The idea is clear and the paper is well-written. The experiments also verify the effectiveness of proposed method.

**Weaknesses:**

I do not witness obvious weakness of this paper.

**Questions:**

It seems that start epoch is a sensitive parameter, which is different in different model. So how is start epoch defined? By grid search?

---

> ### Author Response · Authors · 2023-11-22
> **Response to Reviewer jMFR**
>
> We thank the reviewer for the feedback, for noting that our method is clear and effective, and for finding our paper to be well written.
>
> **It seems that start epoch is a sensitive parameter, which is different in different models. So how is start epoch defined? By grid search?**
>
> Empirically we found that applying AWP works best after CTC training loss reaches 80-90% accuracy, which may vary across different settings (i.e. models, datasets, etc). We elaborate more on the start epoch parameter in the 'common response' part, we kindly refer the reviewer to this part.

---

### Official Review · Reviewer_bLBJ · 2023-11-01

**Soundness:** 3 good
**Presentation:** 3 good
**Contribution:** 3 good
**Rating:** 6
**Confidence:** 4

**Summary:**

For speech recognition task with CTC loss (and others) we assume equal weights between different alignments and we optimize the total probability of all correct alignments. Authors of the paper are concerned about the latter fact and push for considering different weights for different alignments. Authors propose an independent loss, plug and play, which will control what alignments are more preferable to inherit desired properties, like latency (emit tokens faster, without delay and drift) and minimum WER instead of loss. They claim that proposed method is simple compared to prior works and adds few lines of code with no need to change the main loss function (e.g. CTC) used in training. Results on low latency and minimum WER for different models (transformer, conv), data scale and tasks show validity of the proposed idea.

**Strengths:**

- exploring idea on reweighing alignments depending on the task / desired properties, e.g. better latency
- simplicity of the method, including plugin property instead of modifying CTC loss itself

**Weaknesses:**

- absence of comparison of proposed method with prior works and baselines (e.g. if we introduce reweighing of alignments directly in the CTC loss)
- complexity of additional hyper-parameters choice (no robustness), e.g. when we start the additional proposed loss function optimization
- missing details on how exactly the sampling of the word to be corrected is implemented, as it could be that no word is available for substitution. Why WER and not CER as language model could fix the errors? Is language model used in this process?
- I believe that sampling alignments is not the right / optimal way: we could consider optimization of the top-N alignments instead, as otherwise we spend time on optimization low probability alignments.
- A bunch of models in Table 2 (on latency empirical results) are not comparable: either latency should be fixed and WER is compared or vice versa. Right now it is hard to make any conclusions from Table 1 due to different values for latency and WER of different models.
- Results in Table 3 are within std on Librispeech as 0.1 variation is normal between different models often for clean part of test set. Also it is not clear if improvement is consistent for both greedy and LM decoding or only for the latter one. Greedy decoding should be reported too to make full clear picture how the proposed method improves results.
- I found it overall hard to formulate the proper reweighing between different alignments, rather than simple way of controlling the latency by restricting context or optimising directly mWER. It is not clear why proposed way of sampling alignments is sufficient or significantly beneficial. Overall, reported results are only marginal.

**Questions:**

There are many typos in the text, including missing dots in the end of sentences, usage of words with capitalisation for the first letter and some ambiguity in sentence formulation, style of citations in brackets or without brackets, dashes usage. Proof-read is needed for the final revision.

Comments / Questions / Suggestions
- "CTC posteriors tend to be peaky (Zeyer et al., 2021; Tian et al., 2022), and hence the posterior of one specific alignment is dominant over the others." I would smooth this formulation a bit, as likely several tokens are dominant for each time frame, and thus set of alignments (few) are dominant, not only one as discussed in a bit in Likhomanenko, T., Collobert, R., Jaitly, N., & Bengio, S. (2023, February). Continuous Soft Pseudo-Labeling in ASR. In Proceedings on (pp. 66-84). PMLR.
- what will happen if we choose top-N alignments instead of sampling them?
- what will happen if we use several tokens removal for the latency function instead of only 1 token? is it improving latency?
- "35K hours curated from LibriVox" is it multilingual or English only? If multilingual, why not English only as then another confound factor is introduced?
- I would suggest to report results with both greedy decoding and language model, also report both clean and noisy LibriSpeech as a lot of effects are not visible on clean anymore.
- Throughout the text it is not clear where LM is used or not in the reported numbers.
- Seems in Table 2 performance of conformer model is not so good as in the prior paper (3.7 vs 2.0), or check Squeezeformer baselines.
- What is WSR abbreviation (I could not find this notation in the text)? Figure 6 is hard to parse in the current form.
- Why are multilingual wav2vec used for experiments? why not English only?
- I found it surprising that Gumbel softmax with temperature is similar to the standard sampling. Some discussion would be helpful on this topic in the main body, as seems we are very limited with potential improvements if we manipulate with alignments weights.
- what is the relation between Table 2 and Table 4 for the prior methods? I see that Table 4 has better results in WER than in Table 2.
- why are features computed with "32ms window with a stride of 16ms"? This is really very non-standard Mel filter banks extraction for ASR models.

**Update after rebuttal**

Thanks again for clarifications, additional ablations and patience for my response. I had read again the whole discussion with all reviewers as well as had another pass over the final version of the paper.

I think most of the main concerns are resolved now:
- robustness - usage of the proposed loss from 80-90% WER model is a nice and simple empirical rule
- implementation of the property function for min WER - looks good to me in the code
- ablation that beam didn't help totally makes sense to me, though still interesting that sampling is enough (maybe due to peaky CTC distribution and thus it samples from top hypothesis still)
- proper comparison for Table 2 - I think this is mainly resolved for the comparison with prior works, though still I find the baseline Resnet online vs Resnet online + AWP not directly comparable, but Table 8 with different hyper-parameters makes it then clear that AWP either improves both latency and WER or it improves latency for the same WER performance for all models considered.

The remaining concerns are
- improvements for AWP in min WER task is marginal having std 0.03 for clean set (while for latency task AWP really has significant improvement). I still would like to see greedy decoding results for the final version of the paper, as maybe it improves more, but doesn't make significant improvement after beam-search decoding with LM (as LM knowledge is transferred into acoustic model). This can give more insights in future.
- Results with more tokens shifted or more words correction: it doesn't really improve model - this either show that method is limited or that we incorporate all things over the course of the training (which is possible). In future, will be valuable to have some analysis on that to show if this is really limitation of the method or not.

Based on the above, I am raising score from 3 to 6 (marginally above the acceptance threshold) and soundness and contribution from 2 to 3 both, as I believe results on latency improvement are solid enough, though results on min WER are not very supportive (maybe a weaker models are needed as at the limit of 50k-100k hours of labeled data we don't need anything to have very good models) and thus shows that method has limited impact.

---

> ### Author Response · Authors · 2023-11-22
> **Response to Reviewer bLBJ**
>
> We thank the reviewer for the valuable feedback. We did our best to address the concerns raised by the reviewer.
>
> **Absence of comparison with prior works**
>
> For the latency application, we’ve conducted an extensive comparison to prior works. To ensure a fair comparison, we implemented or adjusted public code to fit our setting. Additionally, we enhanced the text with an additional mWER optimization for comparison, we kindly refer the reviewer to the ‘common response’ for details.
>
> **complexity of hyper-parameters**
>
> This work introduces three additional hyperparameters: $\lambda$ and $\alpha$, standard in the context of hinge loss and additional loss term, and start-epoch. Our rationale for the latter is detailed in the 'common response'.
>
> **Missing details on the implementation. Why WER and not CER? Is language model (LM) used?**
> We optimize WER rather than CER as WER is the primary metric for evaluating ASR systems. Furthermore, CTC is more closely related to CER than WER, given its indifference to whether errors are localized within a single word or distributed across multiple words (see fig 2.b in the text). In response to the reviewer’s comment, we added the code for WER optimization to the supplementary material. Concerning the LM, our implementation uses an LM in the beam search decoding (we kindly refer the review to appendix A1).
>
> **Sampling vs. top-N alignments**
>
> Empirically, we found no evidence in favour of top-N alignments. We elaborated on this issue in the ‘common response’, we kindly refer the reviewer to this section.
>
> **Comparing models in table 2**
>
> Each of the top 2 sections in table 2, along with the Conformer section, exhibit comparability. It presents an offline model, an online model and an online model + AWP, and shows clear trend: the offline model is superior in terms of WER but also has the highest TL, the online model improves the TL on the expense of WER, and the online + AWP improves the TL even further in the cost of a WER.
> The third section presents these 3 models, alongside additional losses or input manipulation. We can see that AWP performs better in both DL and WER.
>
> **Results in table 3 are within std on Librispeech. Report results on Greedy Decoding.**
>
> Although not reported, we repeated all experiments with several seeds. Based on this, we found that our results are exceeding 1 std. We didn’t want to overload the paper with results, therefore we reported after beam search decoding, which is not unusual in the area of mWER optimization.
>
> **I found it overall hard to formulate the proper reweighing between different alignments, rather than simple way of controlling the latency by restricting context or optimising directly mWER. Reported results are marginal.**
>
> We will endeavor to respond to the best of our understanding. AWP only requires a definition of a property function, which given an alignment, yields an improved alignment w.r.t to the property. Thus, it does not explicitly mandate the user to formulate a reweighting scheme. As for the advantages of AWP over alternatives - We discuss in the paper that context restriction allows control of the latency, but it also results in a drift (DL) in the online model. With AWP we were able to reduce this drift at the expense of WER. In table 2 we show that AWP can have a negative drift. Overall, we present compelling evidence that AWP can be successfully applied to low latency applications across, and yield moderate gains in WER optimization.
>
> **CTC posteriors**
>
> We agree, text was updated accordingly.
>
> **Shifting several tokens**
>
> We kindly refer the reviewer to the ‘general response’ part, where we elaborated on variations of the property function.
>
> **35K hours curated from LibriVox**
>
> English only.
>
> **LM usage**
>
> LM is used for decoding. We provide details in Appendix A1.
>
> **Conformer Performance**
>
> We use a different decoder and a different LM than the ones used in the Conformer paper. This could explain the difference. We kindly refer the reviewer to the ‘common response’ for details on the Conformer.
>
> **WSR**
>
> WSR is the word success rate. Figure 6 now shows WER.
>
> **multilingual w2v**
>
> We aimed for the base size model, and as far as we know, this model was trained on the largest training set for such a model.
>
> **Gumbel softmax (GS)**
>
> We show that sampling and GS yields similar results. Since softmax is applied to the CTC matrix, the derivatives of the hinge loss affects all the entries, and not just the entries correspond to the sampled alignment, which can explain the effectiveness of the sampling method.
>
> **The relation between Tables 2 and 4**
>
> Both present other applications to mitigate the DL. To compare table 4 to AWP, we see that AWP achieves the best working point in terms of both DL and WER. Other approaches may yield better WER or latency, but not simultaneously both.
>
> **32ms window, stride of 16ms**
>
>  It is optimized for STFT that works in multiples of 2.

---

> > ### Comment · Reviewer_bLBJ · 2023-11-23
> > **Further discussion**
> >
> > Dear authors,
> >
> > Thanks for the comments and clarification as well as for the code supplementary! If you could update the paper version with marking changes you did to be colored in a different manner -- I appreciate! Please find some other comments I have right now.
> >
> > > Although not reported, we repeated all experiments with several seeds. Based on this, we found that our results are exceeding 1 std. We didn’t want to overload the paper with results, therefore we reported after beam search decoding, which is not unusual in the area of mWER optimization.
> >
> > Could you provide the exact typical std value you get? And this should be included in the revision, e.g. in Appendix if you still don't want to overload the main text. But for readers familiar with the dataset 0.1 difference will be insignificant.
> >
> > >> 32ms window, stride of 16ms.
> >
> > > It is optimized for STFT that works in multiples of 2.
> >
> > Could you clarify if this is used for all models / baselines in the paper, or only for your method? (I believe you set this default for all reported models, but want to be sure).
> >
> > > Each of the top 2 sections in table 2, along with the Conformer section, exhibit comparability. It presents an offline model, an online model and an online model + AWP, and shows clear trend: the offline model is superior in terms of WER but also has the highest TL, the online model improves the TL on the expense of WER, and the online + AWP improves the TL even further in the cost of a WER. The third section presents these 3 models, alongside additional losses or input manipulation. We can see that AWP performs better in both DL and WER.
> >
> > Practically, this doesn't say that your method is preferable for use. I think, more valuable to show results will be to fix latency for all models in the same group and show what WER can be achieved. Otherwise, yep, we know that there is the trade off between latency and WER. If you could show that for the significantly improved latency the WER doesn't change or better than prior works which have worse latency (so your degradation of WER for better latency is lower than in prior works) -- this is "no question" result and contribution for me. With the current numbers I still cannot say why your method will be preferable, as I could decrease model size, or restrict look ahead, etc. to improve latency and get worse WER too. I believe your method is very practical and it is just a lack of proper comparison which must strengthen the paper significantly.
> >
> > This is the main concern for me for the paper acceptance.

---

> > > ### Author Response · Authors · 2023-11-23
> > > **Response to Reviewer bLBJ (2)**
> > >
> > > We thank the reviewer again for the additional feedback. We will try to clarify the remains of the concerns raised by the reviewer.
> > >
> > > The pdf revision prior to the last one has the changes highlighted in blue.
> > >
> > > > Could you provide the exact typical std value you get? And this should be included in the revision, e.g. in Appendix if you still don't want to overload the main text. But for readers familiar with the dataset 0.1 difference will be insignificant.
> > >
> > > We are not familiar with the 0.1 difference for the LibriSpeech Test Clean dataset. Based on our experiments with several seeds across several settings, we received that the std was no more than 0.03.
> > >
> > > > Could you clarify if this is used for all models / baselines in the paper, or only for your method? (I believe you set this default for all reported models, but want to be sure).
> > >
> > > All ResNet and Conformer experiments in the paper worked in this same setting, of 32ms window and stride 16ms. This is highly optimized for edge devices. Of Course, w2v works with the raw audio.
> > >
> > >  >  Practically, this doesn't say that your method is preferable for use. I think, more valuable to show results will be to fix latency for all models in the same group and show what WER can be achieved. Otherwise, yep, we know that there is the trade off between latency and WER. If you could show that for the significantly improved latency the WER doesn't change or better than prior works which have worse latency (so your degradation of WER for better latency is lower than in prior works) -- this is "no question" result and contribution for me. With the current numbers I still cannot say why your method will be preferable, as I could decrease model size, or restrict look ahead, etc. to improve latency and get worse WER too. I believe your method is very practical and it is just a lack of proper comparison which must strengthen the paper significantly.
> > >
> > > We our confident with the performance of our method. We agree with the reviewer's comment that there are multiple ways to demonstrate this, in comparison with the other methods. In our previous response we aimed to demonstrate that our method can reach a working point in which both WER and latency surpasses those of the other methods, in both measurements simultaneously. For example, other methods can reach a better WER, but in this case their latency will be larger than ours. If we think of it as a 2 axis graph of latency and WER, then our method reaches the point that is better on WER and latency at the same time. In regard to this working point, other methods will either have both measurement inferior to ours, or one of the measurements could be better, but the other will be worse.
> > >
> > > To specifically address the reviewer's concern, we fixed the WER of the models, as much as we could, and tested their latency. For this, we took the minimal WER that one of the other methods could reach. We tested AWP and the other methods, as much as possible, at this particular WER, and showed that our model has the minimal latency at this working point. We've updated the text to include additional table with this comparison in appendix E.

---

### Official Review · Reviewer_Nofc · 2023-11-11

**Soundness:** 3 good
**Presentation:** 3 good
**Contribution:** 3 good
**Rating:** 6
**Confidence:** 5

**Summary:**

The paper proposes a framework to train a CTC model with a desired property by adding an auxiliary loss. It samples N alignments from a pre-trained CTC model, feeds them to a property-designed function to get N better alignments w.r.t. the property, then adds a hinge loss on each pair of alignments as an auxiliary loss to the original CTC loss, to increase the probability of the better alignments with the desired property. The proposed framework is experimentally tested in applications to optimize latency and WER respectively and has shown improvement of the designed property compared to the vanilla CTC training and some other existing approaches.

**Strengths:**

- The proposed framework is flexible and simple enough to generalize to different properties and provides a generic way to makes the CTC training more controllable.
- For the low latency application, the proposed framework achieves better latency and quality tradeoff than a few existing approaches and be on par with another best approach (TrimTail).
- For the minimum word error rate application, the proposed framework achieves some modest improvement over the MLE baseline.

**Weaknesses:**

- For the minimum word error rate application, there has been a number of work in optimizing it in a discriminative sequence training setting for ASR, e.g. MBR training, large margin training, etc. The proposed framework should be compared to those stronger baselines instead. Right now it is only compared to the weaker vanilla MLE baseline with some modest improvement. In particular, if the property function is to generate the ground truth alignment, instead of only allowing reducing 1 word error, then it should be closer to the traditional discriminative sequence training setup.
- For the minimum word error rate application, Prabhavalkar et al. 2018 found that using the n-best beam search hypotheses is more effective than the sampling-based approach. This paper should do a similar comparison whether it should compute the property function in the n-best alignments instead of sampled alignments.
- Latency and WER optimizations typically compete with each other. It would be great to utilize the proposed framework to optimize these two properties jointly with a single property function to see if they can be balanced better together, and also see how effective and generalizable the framework is.
- How sensitive is the optimization to the specific choice of the property function? E.g. the currently designed latency function only allows one time step faster, and property function for the minimum WER application only allows one word error reduction. Are these choices made in order to stabilize the training, or actually they can be relaxed to allow more changes as well? The paper should compare more different property function choices for the same specific application.
- For the latency application, another intuitive approach would be to sample an alignment corresponding to the ground truth label sequence, and then the property function would be to run a force aligner to get the more accurate time alignment for the label sequence. How would this compare to the current proposed approach?
- Adding the latency optimization to an offline Conformer model doesn't make much sense, since the full-context model is not used in a streaming fashion and it has to wait for the entire sentence to come first, which by itself already has a much higher latency. Conformer can be implemented in a streaming manner as well by just using the left context, which can be optimized for edge devices as well. The latency experiment should be conducted on an online Conformer.
- Using "start epoch" as a tunable hyperparameter to control the balance between latency and quality is a bit strange. How transferable is the optimal start epoch to different learning schedule, model architectures and data?

**Questions:**

- In Figure 6: What is WSR? It is not a standard metric and it is not defined anywhere.

See other questions above in the weaknesses section.

---

> ### Author Response · Authors · 2023-11-22
> **Response to Reviewer Nofc**
>
> We thank the reviewer for the valuable feedback, which assisted us clarify some aspects of the work and update the text accordingly. We are particularly thankful for the recognition of the strengths of our work. In the following, we did our best to address the comments and the concerns raised by the reviewer.
>
> **Comparison to discriminative sequence training (DST)**
>
> We thank the reviewer for the comment. The focus of this work is CTC, and thus we focused on comparing AWP to other optimizations that are suitable for CTC. As noted by reviewer 4pZf, we conducted an extensive comparison in the latency application. For the WER application, to the best of our knowledge, there is a limited body of work on DST with CTC. Graves & Jaitly, 2014 is one example, yet, it required extensive optimization. Thus, we adapted another approach by Prabhavalkar et al,. 2018 to work in a CTC setting. With respect to the reviewer’s suggestion for the property function, we believe it will be an interesting direction to explore and see if it’s closer to other DST baselines. We kindly refer the reviewer to the ‘common response’ for more details.
>
> **N-Best beam search (BS) hypotheses Vs. Sampling**
>
> We thank the reviewer for this comment. We have conducted experiments with both sampling and n-best BS hypotheses, which didn’t indicate that n-best BS is superior to the sampling method. Moreover, running a BS decoder at every training step requires more resources and training time. We kindly refer the reviewer to the ‘common response’ for elaborated explanation and a speculation as to why n-best BS method is not superior in this case.
>
> **Latency  WER optimization compete with each other**
>
> WER and Latency were two property examples for AWP. We optimized each property separately in order to compare with previous results. Yet, the simplicity of AWP allows us to combine the losses together, or to use a single property function that improves both properties simultaneously, and test how the two properties compete with each other. We added this comment to the discussion in section 5.
>
> **Is the optimization sensitive to a specific choice of the property function?**
>
> We experimented with multiple implementations of the property functions, and eventually reported the simplest (and evidently the most effective) implementation. Empirically, we found that other implementations worked as well. We added appendix C in the paper to shed more light on that matter. We kindly refer the reviewer to the ‘common response’ for more details on the different variations of the property function.
>
> **An Alternative approach for the latency application which samples ground truth (GT) alignments and compares them to an alignment obtained by a force aligner**
>
> We presume the objective is to derive a forced alignment from an offline model. Otherwise, drawing it from the concurrently trained online model introduces uncertainty regarding its time accuracy and latency compared to the random GT alignment. While one might expect the online model's latency to match the offline model, negative values in table 2 indicate our property function implementation encourages the online model to drift beyond the offline. Additionally, our implementation diverges from prior works like Tian et al., 2022 by extending beyond perfect alignments. The choice not to limit the property function to perfect alignments aligns with the notion that streaming ASR systems should emit fast even if the transcript is inaccurate.
>
> **AWP with an offline Conformer**
>
> We appreciate the reviewer's feedback. The Conformer experiments were not fully matured by the ICLR deadline. We chose to present the offline model with AWP as a proof of concept, assuming success with an offline model implies potential success with an online model. To adapt the offline model to an online setting, at inference we restricted its context (both right and left) so it works in a streaming manner. We updated table 2 with the online Conformer, and show that although it wasn’t trained as an online model, but inferred as one, the performance of the Conformer with AWP shows a significant DL reduction and only a mild reduction in WER.
>
> **Start Epoch as a tunable hyperparameter for balancing between latency and quality. How transferable is the optimal start epoch to different settings?**
>
> Figure 6 right indicates that tuning alpha is a better practice for balancing between latency and accuracy. However, figure 6 left shows that tuning start-epoch allows some balancing, as there’s a ~11% WER deterioration if applied after 0.9 epochs instead of 16.7. Empirically we found that applying AWP works best after CTC training loss reaches 80-90% accuracy, which may vary across different settings (i.e. dataset, etc). We kindly refer the reviewer to the ‘common response’  for more details.
>
> **WSR in figure 6**
>
>  WSR is word success rate, 100-WER. We changed figure 6 to present WER.

---

> > ### Comment · Reviewer_Nofc · 2023-12-05
> >
> > Thanks for the extra efforts in answering all the questions and adding additional experiments. They in general addressed most of my concerns, with a couple left:
> >
> > - Increasing the number of shifted tokens or corrected words as the property function makes the proposed approach less effective, especially for the latency optimization. It would be good to do more analysis to better understand why this is the case, in the final revision or maybe future work.
> > - The improvement of the latency optimization application is clear compared to the other existing approaches, especially after the authors added extra experiments in Appendix E during the rebuttal. But the improvement of the WER optimization application is marginal compared to MWER_OPT.
> >
> > So I raised my rating to "6: marginally above the acceptance threshold".

---

### Author Response · Authors · 2023-11-22
**Common Response**

We thank the reviewers for their valuable feedback and helpful suggestions. We have updated the text to incorporate these suggestions. We are encouraged to hear that the reviewers recognized the simplicity of the proposed method, as well as its effectiveness. We found some common questions, which we find useful to address commonly.

### Varying the property function

We had extensive research with different property functions. Following the insightful comments of the reviewers, we now include some of these results in the paper in appendix C. Specifically, in tables 4, 5 we show that optimizing AWP with shifting alignments by more than 1 token, or correcting more than 1 word overall followed the same trend. Therefore, the final version of the paper included the simplest (and also most effective) implementation. We conclude that the optimization is robust, to a large degree, to the implementation of the property function.

### More results on the mWER optimization
We appreciate the reviewers’ comments on this subject. In a seminal work, Graves & Jaitly, 2014, suggested a sequence level loss for optimizing the expected WER. However, implementing their loss requires an extensive optimization in the backward pass, based on eq.22, which is in contrast to AWP which is straightforward for implementation. Prabhavalkar et al., 2018 offered a similar approach for expected WER optimization, designed for attention based models trained with the CE loss. We adapted their implementation for CTC, and trained a model with their expected WER loss, similarly to AWP. The results we achieve with their approach are similar to using AWP, emphasizing the general nature of AWP. In response to the reviewers' feedback, we have included a dedicated discussion and the corresponding results in Section 4.2.

### N-best Beam Search Hypotheses Vs. Sampling:
We thank reviewers for the comments. Several studies indicate that using n-best beam search (BS) hypotheses is favorable (Prabhavalkar et al., 2018), and some use sampling (e.g. Graves & Jaitly, 2014, [1] ). Despite conducting experiments with both methods, our empirical findings reveal no evidence in favor of n-best BS hypotheses over sampling in the context of CTC training.
We speculate that BS n-best is more effective in settings other than CTC (e.g. when training Transducers). Since the CTC outputs are independent, applying BS is equivalent to taking the n-best alignments according to eq.1 in our paper. Therefore, these alignments are expected to be very similar to each other, and in most cases will differ by only 1-2 tokens, typically involving a character replaced with a blank or vice versa. Consequently, these alignments share similar transcripts which is less effective for optimizing WER for example. In general, we suspect that similar alignments means they are concentrated in a narrow area in the output space. Alternatively, BS can incorporate a language model (LM), as suggested by Graves & Jaitly, 2014. Then the addition of a language knowledge that is absent in the acoustic outputs (i.e. in the CTC matrix) may push the process towards alignments with low probability. To conclude, obtaining n-best BS hypotheses, with or without an LM, may lead to undesirable results, which may explain our empirical findings. Additionally, running a beam search decoder at every training step consumes more resources and requires more training time.

### The role of the ‘start epoch’ parameter
Previous works by Graves & Jaitly (2014) and Prabhavalkar et al. (2018) proposed incorporating WER optimization with an additional loss into a pre-trained model. However, these works don’t introduce a scheduling for when to apply this loss. In an attempt to design AWP as a general framework, to be applicable to WER optimization among other applications, we found that the start epoch is a key element, and we sought to quantify it. Empirically, we found that applying AWP when the training loss reaches 80-90% accuracy is most useful.
Quantifying it for the latency application shows that while it has a significant impact on the drifting, it has a moderate impact on the WSR (see fig. 6 left).

### AWP with Conformer
We added another entry to table 2 for a Conformer model which was inferred in an online (streaming) fashion. We updated the text to provide the details in sections 3, 4.1 and appendix A.2.

### WSR in figure 6:
WSR is word success rate, 100-WER. We changed figure 6 to WER instead.

[1] H. Sak, M. Shannon, K. Rao, and F. Beaufays, “Recurrent Neural Aligner: An Encoder-Decoder Neural Network Model for Sequence to Sequence Mapping,” in Proc. Interspeech, 2017, 1298–302.

---

### Meta-Review · Area_Chair_cncM · 2023-12-01

**Metareview:**

In this paper the authors propose the so-called Align With Purpose (AWP)  framework which is a regularizer added to the conventional CTC loss to promote alignments with desired properties to specific tasks.  The authors investigate its effectiveness on low latency streaming and minimum word error rate (MWER) training in ASR on three datasets:  the 960-hour Librispeech, the 35k-hour LibriVox and a 280k-hour large scale in-house dataset.  The experimental results appear to be supportive.  There has been some fruitful discussion between the authors and reviewers to clear up most of the concerns raised in the review.  Overall, the work is interesting with solid experimental results. That being said, there are some standing concerns. First of all, the proposed AWP seems to be helpful for low latency training but the improvement is very incremental in MWER even with careful hyper-parameter tuning.  The authors need to provide stronger evidence to show its effectiveness on MWER on more datasets and model architectures. Second, as one reviewer pointed out, it would be interesting to show how well low latency and MWER training would work together as this is a natural question to ask.

**Justification For Why Not Higher Score:**

The proposed AWP is clearly helpful for low latency streaming but the improvement on MWER is not as convincing.

**Justification For Why Not Lower Score:**

This is a solid work in general with good results.

---

### Decision · Program_Chairs · 2024-01-16

Accept (poster)